# Nanostructure of Porous Si and Anodic SiO_2_ Surface Passivation for Improved Efficiency Porous Si Solar Cells

**DOI:** 10.3390/nano11020459

**Published:** 2021-02-11

**Authors:** Panus Sundarapura, Xiao-Mei Zhang, Ryoji Yogai, Kazuki Murakami, Alain Fave, Manabu Ihara

**Affiliations:** 1Department of Chemical Science and Engineering, Tokyo Institute of Technology, 2-12-1 Ookayama, Meguro, Tokyo 152-8552, Japan; sundarapura.p.aa@m.titech.ac.jp; 2Department of Mechanical Engineering, Tokyo Institute of Technology, 2-12-1 Ookayama, Meguro, Tokyo 152-8552, Japan; 3Department of Chemistry, Tokyo Institute of Technology, 2-12-1 Ookayama, Meguro, Tokyo 152-8552, Japan; energy.power.ry.516@gmail.com (R.Y.); kazuki.techtech@gmail.com (K.M.); 4Univ Lyon, INSA Lyon, ECL, CNRS, UCBL, CPE Lyon, INL, UMR5270, 69621 Villeurbanne, France; alain.fave@insa-lyon.fr

**Keywords:** solar cell, porous silicon, surface passivation, anodic oxidation

## Abstract

The photovoltaic effect in the anodic formation of silicon dioxide (SiO_2_) on porous silicon (PS) surfaces was investigated toward developing a potential passivation technique to achieve high efficiency nanostructured Si solar cells. The PS layers were prepared by electrochemical anodization in hydrofluoric acid (HF) containing electrolyte. An anodic SiO_2_ layer was formed on the PS surface via a bottom-up anodization mechanism in HCl/H_2_O solution at room temperature. The thickness of the oxide layer for surface passivation was precisely controlled by adjusting the anodizing current density and the passivation time, for optimal oxidation on the PS layer while maintaining its original nanostructure. HRTEM characterization of the microstructure of the PS layer confirms an atomic lattice matching at the PS/Si interface. The dependence of photovoltaic performance, series resistance, and shunt resistance on passivation time was examined. Due to sufficient passivation on the PS surface, a sample with anodization duration of 30 s achieved the best conversion efficiency of 10.7%. The external quantum efficiency (EQE) and internal quantum efficiency (IQE) indicate a significant decrease in reflectivity due to the PS anti-reflection property and indicate superior performance due to SiO_2_ surface passivation. In conclusion, the surface of PS solar cells could be successfully passivated by electrochemical anodization.

## 1. Introduction

Solar cells, as a renewable energy source, will significantly contribute to clean energy, which is an energy goal that has become one of the greatest challenges to fuel our economies. For solar cells, silicon (Si) is a promising material that combines suitable optoelectronic properties with earth-abundance and technological availability [1]. Si solar cells are dominating the photovoltaic (PV) market, in which state-of-the-art *c*-Si solar cells can achieve about 26.7% efficiency [2], approaching the Shockley–Queisser limit of 30% [3]. In order to surpass the theoretical efficiency limit for single junction solar cells, a tandem structure is a potential candidate. By selecting the material based on the band gap energy by considering the atomic lattice matching, nanostructured silicon material could be a promising candidate for the top layer of the tandem structure. For decades, there has been extensive research into potential candidates for the top layer, and the most prominent Si-based materials are Si nanowire [4] and nanopore arrays [5,6]. These structures are composed of an array of vertically aligned Si nanostructures, which can not only reduce the volume of semiconductor absorber but can provide nearly ideal photon absorption across the solar spectrum for high efficiency. When the pore size of the surface features equals or exceeds the wavelength of the incident light, nanostructures provide beneficial light-trapping that increases the effective light-path length in the Si [7,8]. When its porosity is increased, porous Si (PS) exhibits decreased reflectance (3~30%) and increased band gap (1.1~1.9 eV) [9]. Such unique features make PS a promising material for use in solar cell technology and an interesting candidate for the top absorber in Si tandem cell applications [10].

As high aspect-ratio nanostructures, like porous silicon, are prone to an uncontrollable oxidation of native oxide when exposed to the atmosphere, many studies have been conducted on various passivation techniques to improve their porous surface stability, e.g., wet thermal oxidation [11], rapid thermal treatment in NH_3_ [12], and room temperature oxidation with ozone [13]. However, as these techniques only applied to the PS layer without *p-n* junction, their applications are only limited to PS as the photoluminescence (PL) emitting layer for other applications, like biosensors and microelectronics. In terms of applications in PV cells, compared with conventional cells in the present time, solar cells based on nanostructured Si [14,15,16] exhibit lower efficiencies without surface passivation. The main challenges that hamper the application of PS in solar cells are related to the increased photocarriers at the dramatically increased surface area of nano-porous Si causing reductions in both the device’s short-circuit current and open-circuit voltage, and consequently causing a decrease in the efficiency (*η)*. This indicates that the passivation of PS surface is important to improve the performance of the nanostructured solar cell.

In terms of passivation of the surface of Si, Al_2_O_3_ films deposited on Si surfaces via atomic layer deposition (ALD) possess negative fixed charges at the Al_2_O_3_/Si interface, and thus provide superior passivation quality for *p^+^* emitters of *n*-type-based solar cells [17,18]. For phosphorus diffused *n^+^* emitters of *p*-type solar cells, SiO_2_ films can provide good passivation due to their excellent chemical passivation for the Si-SiO_2_ interface [19]. The recent study by Grant et al. demonstrates that the thin SiO_2_ film (~70 nm) formed by the anodic oxidation method at room temperature can exhibit an excellent surface passivation quality and durability on the *c*-Si substrate with a high *ƞ* of >23% [20]. However, the anodic oxidation was not applied to the nanostructured solar cell. In the case of nanostructured solar cells, additional technological issues, such as the controllability of SiO_2_ thickness, the effectiveness of passivation by the nanometer scale thin SiO_2_ layer, and the preservation of the original structure, must be developed. Oh et al. fabricated a *p*-type-based black Si solar cell in which a thermal SiO_2_ passivation layer provided a very low effective surface recombination velocity of 53 cm s^−1^ on the surface and yielded an *ƞ* of 18.2% [21]. However, the process to thermally grow SiO_2_, despite having a small effect on the structure when applied to *c*-Si, has a limitation when applied to PS nanostructures that have an aspect ratio as high as 400:1 [22]. The high operating temperature of thermal SiO_2_ deposition at 400–1100 °C [11,12] will damage the nano-porous layer [23]. Therefore, the method to deposit an oxide passivation layer on nanostructured Si must be designed in such a way that it can precisely control the thickness of the oxide to prevent a complete oxidation, while the processing temperature must be low enough to preserve the original nanostructure.

In this work, we used a simple, low-cost method of electrochemical etching to prepare PS layers with a *p-n* junction with low reflectivity on *n*-Si emitters of Si solar cells. The PS was composed of crystalline nano-pillars and possessed low reflectivity in the ultraviolet (UV) and visible regimes (300–1100 nm). Electrochemical anodization was applied to passivate the surface of the PS layer because this passivation technique allows the precise control of the oxide layer thickness and can be applied at room temperature in a short processing time, which also reduces the risk of adversely affecting the original nanostructure of PS compared to conventional high temperature passivation processes. This low-cost approach combines both the good surface uniformity of SiO_2_ and the effective oxidation of the dangling bond interface states to provide sufficient passivation for PS solar cells. These PS solar cells could efficiently reduce the recombination at the front surface via a bottom-up anodization mechanism used to form the SiO_2_ passivation layer on the surface of the pores. The effect of the anodization duration on the main features of the PV devices was demonstrated systematically by evaluating the PV performance of these anodic SiO_2_ passivated PS solar cells. In these PS solar cells, an optimum anodization duration of 30 s produced a maximum *ƞ* of 10.7%, associated with passivation time (*t*) effects on the open-circuit voltage (*V_oc_*), short-circuit current density (*J_sc_*), fill factor (*FF*), shunt resistance (*R_sh_*), and series resistance (*R_s_*). The functionality of the PS layer for improving the PV performance of solar cells was investigated here by varying the SiO_2_ layer thickness.

With the existing technologies to fabricate the PS layer in more scalable production [24], e.g., in microelectronics [25] and high-volume industrial microelectromechanical systems (MEMS) production [26], together with the reduction in the manufacturing cost by the double layer transfer process (DLPS) method that allowed the utilization of the silicon wafers in a more efficient way [6,27,28], the fabrication of PS as well as the SiO_2_ passivation could implement these technologies, with some modifications, in the actual PV production scale. Therefore, as the critical step to develop a more feasible passivation technique for PS solar cells, the electrochemical passivation of PS as the top emitter of Si solar cell at room temperature highlights a promising possibility that this low-cost process for PS structure can improve the light-trapping in thin-film Si solar cells and possibly be used as a wide band gap, top-cell material in Si tandem cell applications in the next generation solar cells.

## 2. Materials and Methods

The substrates of the PS solar cells fabricated here were *p*-type *c*-Si wafers with (100) orientation, a resistivity of 1–2 Ω cm, a thickness of 200 μm and an area of about 177 cm^2^. The Si wafers were ultrasonically cleaned and then dipped in dilute HF 1% (Wako Pure Chemical Industries, Ltd., Osaka, Japan) solution to remove the native oxide. A phosphorus (P) doped *n* layer was deposited on the *p*-Si wafer surface via spin-on doping (SOD Standard Solution P8545PV, Filmtronics, Butler, PA, USA) with post-annealing at 900 °C (MILA-5000 series, Advance Riko, Inc., Yokohama, Japan) for 17 min under a nitrogen atmosphere. To characterize the doping profile and the *p-n* interface of this fabrication condition, the electrochemical capacitance voltage (ECV) (WEP Wafer Profiler CVP21, Furtwangen, Germany) was measured for four samples (Samples A,B,C,D; Table 1) of phosphorus (P) doped bulk *p*-Si wafers after removal of the thermal oxide from the wafer surface. Repeated measurements were performed for three samples that had a 5 mm^2^ etching area (Samples A-C) and for three different areas (01–03) on the same sample (D). Note that this etching process mentioned in the “etching area” is an electrolytic etching of the doped *c*-Si only for profiling the carrier concentration in each depth from the surface, and differs from the process of fabricating the PS layer in the later section.

To fabricate the PS layer, an anodization process was performed using an in-house constructed wet-etching cell, as shown in Figure 1a, with horizontal orientation of the Si wafers [6,29]. The lower electrical contact to the wafers was provided via a copper base. A cylindrical cell containing the electrolyte of 46% HF: 99.5% ethanol = 1:1 solution was placed on top of the Si wafer. The surface area size of the PS layer formed on the Si wafer surface was controlled using an O-ring with an inner diameter of 20 mm. The cathode was a platinum net (15 mm in diameter) immersed in the electrolyte. Etching was performed under dark conditions at room temperature. Furthermore, for the samples with surface passivation, the formation of anodic SiO_2_ was carried out by in-situ surface passivation of the PS layer using an anodization process in a HCl/H_2_O solution at a constant current of 0.36 mA/cm^2^ for different passivation time *t* from 6 s to 135 s.

With this fabrication condition, PS layers of different thicknesses and an average porosity of 38% were prepared on *c*-Si substrates. A porosity of 38% was calculated based on the effective medium approximation (EMA) with Bruggeman’s Model [30] and is optimal to achieve a low lattice-mismatch between the PS layers and *c*-Si substrate. For the surface passivation, a thin SiO_2_ film was formed to cover the surfaces of the pores on the PS layer by the anodization method in H_2_O/HCl at ambient temperature. The anodization reaction is as follows:Si + 2H_2_O ↔ SiO_2_ + 2H_2_(1)

Based on this porosity and the kinetic of the anodization reaction (1) [31], the volume ratio of SiO_2_ thin film per volume of PS layer in each *t* can be calculated, so that it can be confirmed that the PS layer is not completely oxidized, i.e., the volume ratio is equal to 100%. Table 1 summarizes the preparation conditions and layer thickness of all the samples fabricated for this study.

Transmission electron microscopy (TEM, 200 KV JEM-2010F, JEOL, Tokyo, Japan) was used to investigate the microstructures of the PS-1 layer, PS-2 layer and PS-3 layer. TEM specimens were prepared using a focus ion beam (FIB) (FB-2100) procedure. A protective 200-nm-thick coating of carbon was deposited on the surface of the PS layer. A foil from the PS layer was obtained using Ga+ ions beam to excavate the material from both sides to a depth of 5 μm. Finally, the foil was extracted from the PS layer and deposited onto the TEM membrane. The morphology of the PS cells was characterized using a Field Emission Scanning Electron Microscope (FESEM, JSM-6301FZ, JEOL, Tokyo, Japan). FESEM samples were taken from cut samples of the PS-2 cell and passivated PS-2 cells that had been Pt coated (Platinum Auto Fine Coater JFC-1600, JEOL, Tokyo, Japan) for better conductivity in this FESEM imaging.

Finally, to evaluate the PV performance of the fabricated cells, for an active area of 1 cm^2^ on the as-fabricated PS-1 cell, passivated PS-1 cell, PS-2 cell, and passivated PS-2-(a-h) cells, the contact electrodes (50-nm-thick Ag films) were deposited with in-house resistive thermal evaporator onto both the front and rear surfaces, as shown in Figure 1b. A fishbone-shaped electrode structure with a 0.39 cm^2^ area was used for the front contact. Note that the thin Ag electrode in this experiment has a low transmittance on the wavelength of 300–1200 nm, which is the range that *c*-Si solar cell and PS solar cell absorb the light. Thus, the PV parameter can be calculated based on only the active area. In addition, because it is needed to evaluate only the true effectiveness of the anodic SiO_2_ passivation layer, this fabrication method was designed to be as simple as possible, e.g., without back surface field (BSF). The structure of the PS solar cell would be as follows (from top to bottom):

Ag front electrode/anodic SiO_2_/N+ emitter PS/ P-type *c*-Si/Ag back electrode.

Then, the current density–voltage (*J–V*) curve and other PV parameters (*J_sc_*_,_
*V_oc_*, *FF*, and *ƞ*) were obtained using a solar simulator (CEP-25MLH Spectrometer, Bunkoukeiki, Tokyo, Japan) under AM1.5 illumination (Xenon lamp model XCS-150A, JASCO, Tokyo, Japan). The EQE data were obtained using a spectral response measurement system (CEP-2000ML, Bunkoukeiki, Tokyo, Japan), while the IQE data were calculated based on the reflectance data obtained using UV-Vis spectrophotometers (V-570, JASCO, Tokyo, Japan). Note that the *J_sc_*, *ƞ*, EQE and IQE shown in the Results and Discussion section were calculated based on the active area of the solar cell surface. As reference, *c*-Si solar cells were fabricated under the same conditions as those for PS solar cells.

## 3. Results and Discussion

### 3.1. Characterization of Nanostructured PS and Depth of P-N Interface

To investigate the depth of the *p-n* interface, ECV profiling was performed on phosphorus (P) doped bulk *p*-Si wafers after removal of the thermal oxide from the wafer surface (Samples A,B,C,D; Table 1), to obtain the activated P concentration profiles at 900 °C for 17 min.

In both the arithmetic scale and logarithmic scale (Figure 2), the profiles of these samples show a steep drop-off in the carrier concentration around the wafer depth of 200 nm, which indicates a high P-doping concentration of the *n*-layer on the top of the *p*-Si wafer. Despite the difference of the carrier concentration between samples from the top surface to around 150 nm depth (Figure 2a), it does not significantly affect the position of the *p-n* junction, as the difference in this highly doped sample is almost similar, as can be seen in the logarithmic scale (Figure 2b), and the carrier concentration drastically drops down in the same manner in every sample. Repeated measurements from different samples (A–D) and from areas 01–03 from sample D confirm that a uniform *n*-layer had formed on the top of *p*-Si wafer by thermal diffusion of the P-dopant. Figure 2c shows a cross-sectional TEM image of the 545-nm-thick PS-3 layer on a *c*-Si substrate, revealing bright and dark contrast layers due to their different porosity. The porosity in the PS-3 layer formed on the top bright layer (200-nm-thick) differed from that in the PS-3 layer on the bottom dark layer (270-nm-thick), indicating a different P-doping density of the Si *n*-type region. The porosity of PS can, therefore, be controlled by adjusting the P-doping level [32]. Note that the depth of the top bright contrast layer coincides with the high P-doping concentration of the *n-* layer indicated in the ECV profile.

Figure 3a shows an overview of the PS-1 layer with a uniform thickness of 146 nm on a *c*-Si substrate. The pores appear as an amorphous material due to amorphization of the thin Si walls around the pores. The thinness of the Si walls was caused by FIB milling used to prepare the sample for TEM imaging. The projection of the Si skeleton is similar in appearance to a crystalline material. The PS-1 layer exhibits straight pores grown perpendicular to the surface. Uniform contrast in the PS-1 layer indicates uniform porosity, which in turn indicates a uniform doping concentration of the *n*-layer. A high resolution TEM (HRTEM) image of the PS-1/*c*-Si interface is shown in Figure 3b. The crystalline skeleton of the *n*-PS-1 layer has the same orientation as the *c*-Si substrate, indicating an atomic lattice matching (marked by red dotted lines in Figure 3b) at the PS-1/Si interface for homojunction Si tandem-cell applications. Figure 3c shows the 176-nm-thick PS-2 layer. This observed layer thickness (176 nm) is less than the as-fabricated 200-nm-thick PS-2 layer due to a slight top-surface etching effect from the FIB milling pretreatment process for TEM imaging. The corresponding HRTEM image (Figure 3d) shows a smooth PS-2/*c*-Si interface. The speckled contrast of dark and bright regions represents Si crystallites and pores, respectively. In the top PS-2 layer, the pore size is about 10 nm with a mean distance between neighboring pores of about 15 nm, while the bottom PS-2 layer shows a smaller pore size of 5 nm with a neighboring distance of 8 nm.

### 3.2. Electrochemical Passivation Effect in Solar Cells with Thin PS Layer of 146 nm

The reflectivity spectrum for the PS-1 sample with the thin 146-nm PS layer (Figure 4a) shows a significantly lower reflectivity compared to that of the bulk Si sample in the wavelength range of 300–1100 nm measured using a UV-Vis spectrophotometer. For this PS-1 sample, the lowest reflectivity of approximately 9% was achieved for the PS-1 layer at a wavelength of 700–1100 nm, compared to 18% at 450–700 nm. This low reflectivity demonstrates the potential anti-reflection property of the PS-1 porous layer. Figure 4b shows the *J–V* characteristics for this PS-1 cell, passivated PS-1 cell, and the reference *c*-Si cell, measured under AM1.5 illumination. Note that there was no other anti-reflection coating for any of the solar cells in this study. Table 2 lists the measured PV parameters (*J_sc_*, *V_oc_, FF*, *η*) for these three samples. Compared to the *J_sc_* for the *c*-Si cell (23.7 mA/cm^2^), *J_sc_* for the PS-1 cell improved to 27.5 mA/cm^2^. This improvement indicates an increase in the incident photon flux at the *p-n* junction. Figure 4c shows the EQE of these three solar cells. Compared to the *c*-Si cell, the PS-1 cell showed a broad spectral region of photosensitivity to a UV range from 300 nm to 450 nm, suggesting a possible wide optical window due to the PS-1 layer. The overall enhanced EQE for the passivated PS-1 cell is due to the reduction in surface recombination, resulting in a higher *J_sc_* of 30.5 mA/cm^2^ and *FF* of 0.550. The *ƞ* for the PS solar cells with surface passivation increased from 7.93 to 8.54%.

To investigate surface passivation, as shown in the anodization reaction (1), H_2_O oxidation kinetics suggest that no chlorine species is incorporated in the oxide during HCl/H_2_O oxidation [31], while the addition of HCl to H_2_O oxidation ambient could provide uniform oxides with few defects [33]. Moreover, the HCl addition decreases the oxidation rate, which would benefit diffusion control during oxidation of a high aspect-ratio PS layer.

A promising technique for silicon-on-insulator (SOI) technology is oxidized PS at any depth [34]. The bottom-up PS oxidation process is shown schematically in Figure 5a. The PS oxidation rate depends on the density and surface area of the pores of the PS. Due to their differences in density and pore microstructure, the *p*-layer below the *n*-layer is preferentially anodized compared with the *n*-layer Si [31]. The pores of PS can be considered trenches, although anodization reactants travel down from the *n* layer to the *p*-layer. This bottom-up anodization process produces a uniform, broad coverage of oxidation on pore surfaces.

### 3.3. Electrochemical Passivation Effect in Solar Cells with Thick PS Layer of 191–255 nm

These same etching and passivation processes were used here to fabricate PS-2 layers with thickness of 191–255 nm, and which were then surface-passivated via anodic SiO_2_ formation for PS-2 solar cell applications. The uniform morphology of these layers is shown in cross-sectional SEM images in Figure 5b–j. The SiO_2_ anodization process was carried out for passivation of the PS-2 layers for different *t* = 6 s, 10 s, 15 s, 20 s, 30 s, 60 s, 90 s, and 135 s. This scale of SEM images confirmed that this passivation method does not affect the original structure, as evidenced by the uniformity of the porous structure of these passivated samples (Figure 5b–j) that conforms with that of the sample without passivation (Figure 5a).

Figure 6a shows the *J–V* curves under AM1.5 illumination for the PS-2 cell, PS-2-(a-h) cells passivated with anodic SiO_2_, and for the reference *c*-Si solar cell. Table 3 lists the measured PV parameters for these samples. All the cells showed good diode-like behavior, although the effect of *t* is striking. The PS-2 cell (i.e., no passivation) achieved an ƞ of 5.15%, whereas the passivated PS-2-e cell (*t* = 30 s) exhibited the highest *ƞ* of 10.7% (Figure 6b and Table 3). From Figure 6c, although all the cells with a PS layer (data marked with red symbols) show a significant increase in *J_sc_* compared to the reference c-Si cell, which indicates improvement due to the anti-reflection property of a PS layer, the passivation effect is not responsible for this improvement in *ƞ*, as evidenced by *J_sc_* being similar in both the PS-2-e cell (with passivation) and the PS-2 cell (no passivation), 29.8 mA/cm^2^ and 29.9 mA/cm^2^, respectively. The factor that causes this large increase in *ƞ* is due to a significant increase in *FF* from 0.321 to 0.685. The *t* dependence of *FF* is shown in Figure 6d. For passivated cells (PS-2-(a-h) cells)*, FF* increased with increasing *t* up to 30 s and peaked at 0.685. This is attributed to an increase in *J_sc_* without significant loss in *V_oc_* (Figure 6e). The initial increase in *J_sc_* up to *t* of about 30 s indicates a suitable coverage of SiO_2_ film on the surfaces of the pores resulting in the best *ƞ*. These results demonstrate that the bottom-up anodization process successfully fills tiny pores in the PS, and significantly decreases the surface recombination of the PS layer. Longer *t* (>30 s) causes a decrease in *J_sc_* due to lower conductivity from SiO_2_ caused by overoxidation. As shown in Figure 6e, all the PS-2 solar cells had similar *V_oc_*, around 0.5 V, which is lower than the 0.7 V usually obtained from the heterojunction with intrinsic thin layer (HIT) solar cells with a-Si as emitter. This lower *V_oc_* might be due to contact resistance losses at the interface between the Si solar cell and the metal contact and to the impact of electrode geometry.

Figure 7 shows the EQE and IQE for the passivated PS-2 solar cells fabricated for different *t* compared with an as-fabricated PS-2 solar cell (t = 0 s) and a reference *c*-Si solar cell. Although the IQE of solar cells is typically calculated from the ratio of its EQE and spectral absorptance as IQE = EQE/(1-reflectance-transmittance), the IQE spectra shown in Figure 7b were calculated by using the reflectivity of the bare surface without a front electrode.

Throughout most of the spectra at wavelength range of 300–1100 nm, the passivated PS-2 solar cells exhibited enhanced EQE compared to that of the *c*-Si solar cell, attributed to the PS structure with its unique anti-reflection property. Among all the passivated cells, a notable enhancement in the wavelength range of 700–1100 nm occurred when *t* = 30 s. This indicates a long diffusion length due to sufficient surface passivation and lower reflectivity at that wavelength range (700–1100 nm), which consequently yields the highest *FF*. The increase in incident photons trapped at that wavelength range (700–1100 nm) is required for application in thin solar cells because most of the light in the long wavelength region (700–1100 nm) traverses through the semiconductor material unabsorbed. Compared to the passivated PS-2 cells and *c*-Si cells, the as-fabricated PS-2 cell exhibited the lowest IQE throughout the spectra. This low IQE can be attributed to the front surface recombination at short wavelengths and to the rear recombination. The rear recombination can cause a reduction in the absorption of photons with long wavelengths and low diffusion lengths. The low IQE caused by surface recombination eliminates the advantage of low reflectance. IQE values were significantly improved with the anodic SiO_2_ surface passivation (Figure 7b). In particular, a broader spectrum in the wavelength range of 300–1100 nm occurred when *t* =30 s, indicating a sufficient reduction in surface recombination and improvement in minority carrier diffusion length of the passivated PS-2 cells. Currently, ALD films that are 5, 10, or 20 nm-thick are used for passivation of PS-2 solar cells. However, under AM1.5 illumination, these PS-2 solar cells exhibit unsatisfactory PV parameters, which is attributed to the growth of Al_2_O_3_ films being diffusion-limited in these high aspect-ratio PS structures [35]. The consequently downward trend in the saturation front process prevents the reactants from penetrating deep along the pore walls. This is further evidence that for PS solar cells, the anodization SiO_2_ process is more feasible than that of ALD in surface passivation. In contrast, the dominant dependence of the PS-2 thickness on the performance of such solar cells could not be observed (Appendix A).

Figure 8 shows the *R_s_* (8a) and *R_sh_* (8b) of the as-fabricated PS-2 cell (no passivation, *t* = 0), passivated PS-2-(a-h) solar cells as a function of *t*, with the *c*-Si cell as a reference. Compared to the *c*-Si cell, the as-fabricated PS-2 cell exhibits a relatively high value of *R_s_* = 18 Ω cm^−1^, indicating an insufficient movement of light-generated current through the as-fabricated PS-2 layer and the base of the solar cell. More notably, it also corresponds to the lowest *R_sh_* of 0.067 kΩ cm^−1^ in the as-fabricated PS-2 cell, due to the power losses in the as-fabricated PS-2 cell caused by the alternate current path to the light-generated current. Because the as-fabricated PS-2 cell did not undergo passivation treatment, a non-uniform oxide layer caused by the native oxide was formed on the PS layer surface. This nonuniformity of the oxide layer acts as a crystal defect that provides an excessive amount of trapping states, which consequently promotes the recombination process. When this recombination current is strong enough, it can act as a shunt [36]. Therefore, despite having a high *J_sc_* of 29.9 mA/cm^2^ and *V_oc_* of 0.539 V similar to the passivated PS-2 cells (Table 3), a significant amount of light-generated current was lost in the PS-2 cell through the high recombination process due to the nonuniformly passivated surface, thus resulting in the low *R_sh_*, *FF*, and *ƞ*.

For the passivated PS-2-(a-h) solar cells, *R_s_* decreased with increasing *t* from 0 s to 30 s, and then increased gradually, as shown in Figure 8a. At *t =* 30 s, the passivated PS-2-e cell reached the lowest *R_s_* of 2.79 Ω cm^−1^. This is attributed to the enhanced light-generated current through the passivated PS-2-e cell, which had less defect-trapping states as well as less Ag front electrode penetration into the insufficiently passivated pore, and thus leading to a decrease in *R_s_*. However, the additional deposited anodic SiO_2_ (at *t* > 30 s) might turn into an inefficient carrier transport medium. Because the Ag electrode was directly deposited on the passivated PS-2 surface, it might increase the contact resistance between the metal contact and the silicon material. Power losses caused by *R_sh_* represent a parallel high-conductivity path across the *p-n* junction for light-generated current in solar cells. The variation in *R_sh_* with *t* is shown in Figure 8b. The *R_sh_* initially increased rapidly, then peaked at 3.45 kΩ cm^−1^ at *t* = 30 s. This indicates less loss of light-generated current across the *p-n* junction due to the reduction in recombination by surface passivation of the PS-2 layer. The decrease in *R_sh_* with a further increase in *t* corresponds to the increased tendency of *R_s_* at the same *t* conditions. At *t* > 30 s, *R_s_* in the series circuit of solar cells increases due to the overoxidation of anodic SiO_2_ deposition between the metal contact and the silicon. Simultaneously, due to an as-yet unknown cause, the recombination rate is increased when the PS layer is passivated for longer than 30 s. A possible mechanism that decreases *R_sh_* might originate from the crystal defects resulting from overoxidation [36].

## 4. Conclusions

In summary, the cross-sectional view of the microstructure of fabricated 146-nm-thick PS-1 and 176-nm-thick PS-2 layers confirmed a lattice matching at the PS/*c*-Si interface. The lowest reflectivity of approximately 9% was achieved for this thin PS-1 at a wavelength of 700–1100 nm, compared to 18% at 450–700 nm. This highlights the potential of the PS-1 layer as an anti-reflection coating structure in Si solar cells. The broader EQE spectral region of photosensitivity of a passivated PS-1 cell compared to *c*-Si cells at short wavelength suggests a possible wide optical window role of the PS-1 layer.

A bottom-up anodization process successfully produced a uniform SiO_2_ passivation layer with high coverage on high aspect-ratio PS-2 layers without disturbing the original structure. A maximum efficiency of 10.7% with the best *J–V* behavior yielded an *FF* of 0.685 for the PS-2 cell at an optimized passivation time *t* of 30 s (PS-2-e cell). Simultaneously, this PS-2 cell had the lowest *R_s_* (2.79 Ω cm^−1^) and also had the highest *R_sh_* (3.45 kΩ cm^−1^). Such improved PV parameters confirm that the anodic SiO_2_ provides a sufficient passivation approach, especially for nanostructuring a PS solar cell. Notable enhancement in EQE in the wavelength range of 700 to 1100 nm improves the light-trapping at long wavelength (700–1100 nm), and thus provides a promising possibility for improving the light absorption in thin-film Si solar cells. Furthermore, the electrochemical anodic oxidation at room temperature can be used as a surface passivation technique for other nanostructured materials because the thickness of the oxidized layer can be precisely controlled by adjusting the combination of current density and treatment period while maintaining the original nanostructure.

## Figures and Tables

**Figure 1 nanomaterials-11-00459-f001:**
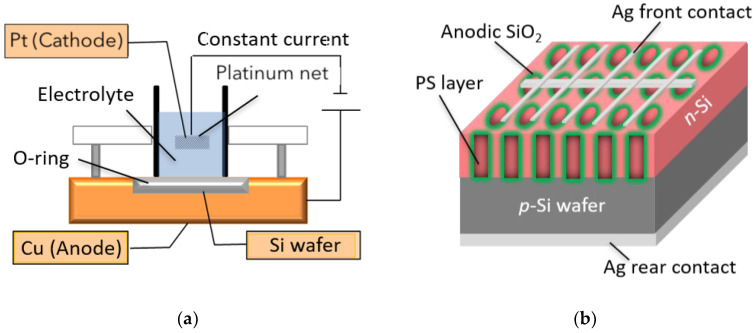
Schematic of (**a**) electrochemical etching set-up; (**b**) solar cells based on porous silicon (PS).

**Figure 2 nanomaterials-11-00459-f002:**
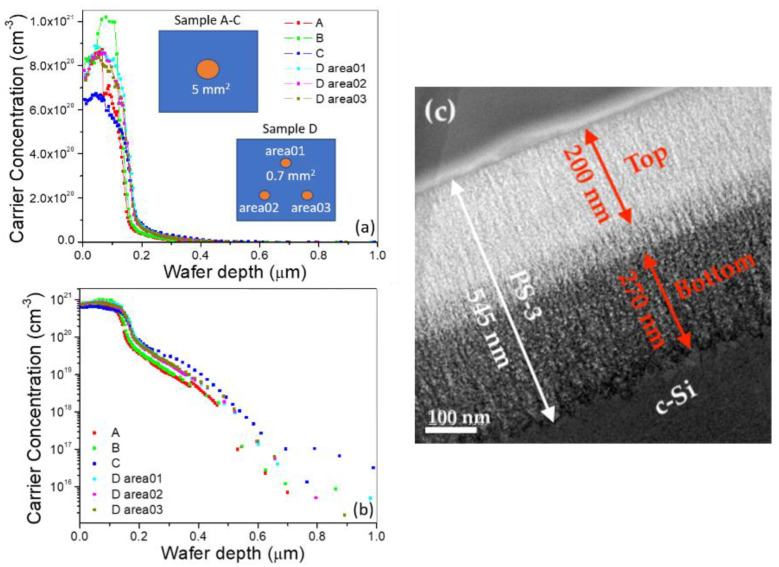
Carrier density dependence on wafer depth by ECV profiling on phosphorus (P) doped bulk Si Samples A–C (5 mm^2^ etching size) and Sample D areas 01–03 (0.7 mm^2^ etching size): (**a**) arithmetic scale; (**b**) logarithmic scale; (**c**) cross-sectional TEM image of 545-nm-thick PS-3 layer.

**Figure 3 nanomaterials-11-00459-f003:**
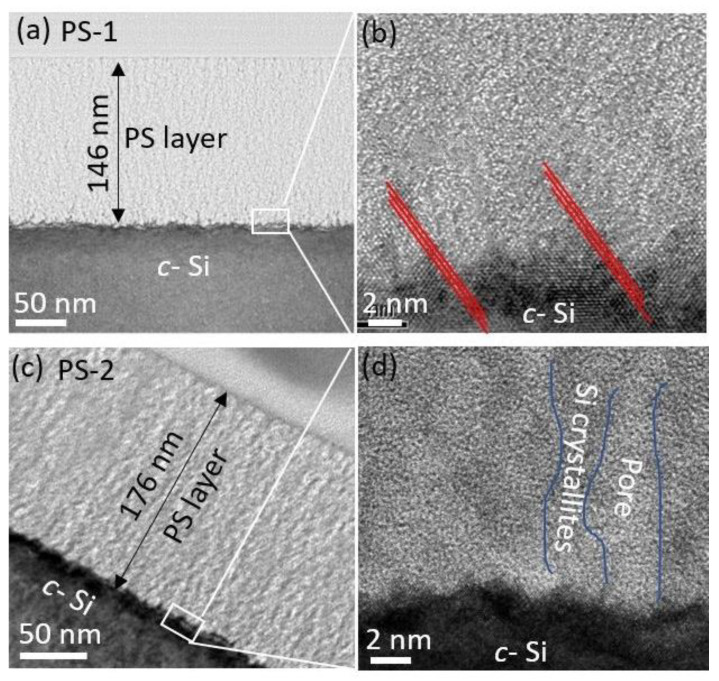
(**a**) TEM image of 146-nm-thick PS-1 layer; (**b**) HRTEM of PS-1/*c*-Si interface at the bottom of PS-1 layer; (**c**) TEM image of 176-nm-thick PS-2 layer; (**d**) HRTEM of PS-2/*c*-Si interface at the bottom of PS-2 layer.

**Figure 4 nanomaterials-11-00459-f004:**
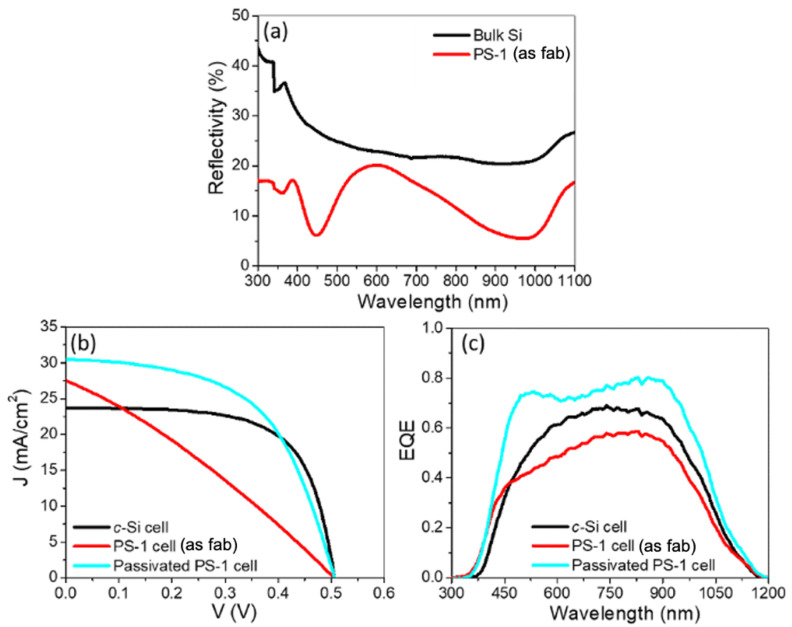
PV characteristics of *c*-Si cell, as-fabricated PS-1 cell and passivated PS-1 cell: (**a**) reflectivity spectra measured for bulk Si and PS-1 layer; (**b**) *J–V* curve; (**c**) EQE spectra.

**Figure 5 nanomaterials-11-00459-f005:**
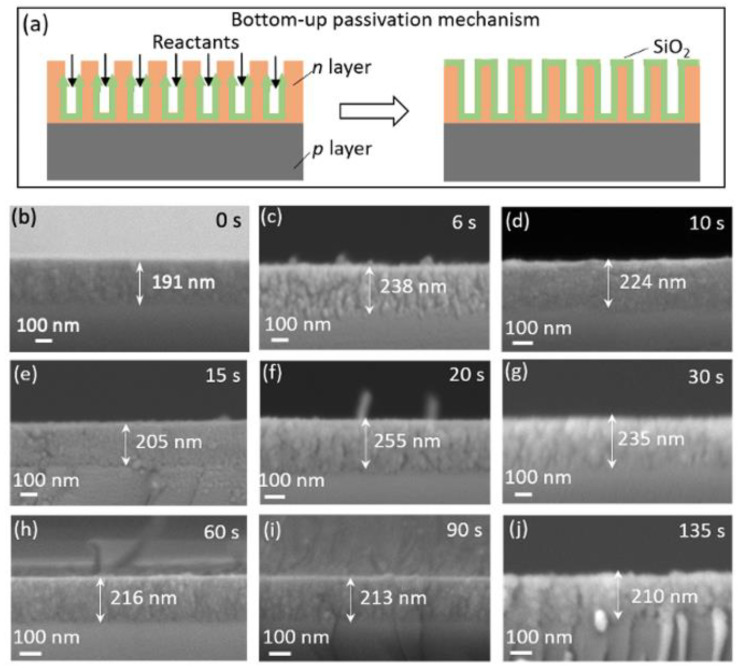
(**a**) Schematic of the bottom-up anodization process of SiO_2_ passivated PS-2 layer; (**b**–**j**) cross-sectional SEM images of fabricated PS-2 layers of different thickness (191–255 nm).

**Figure 6 nanomaterials-11-00459-f006:**
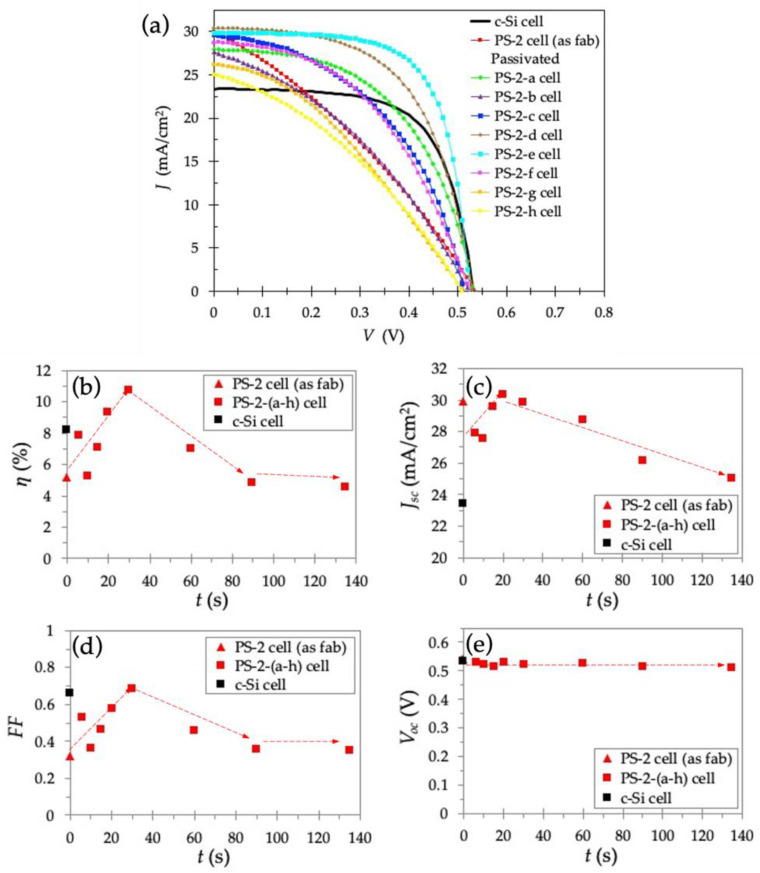
(**a**) *J–V* curves; (**b**) *η*; (**c**) *J_sc_*; (**d**) *FF*; (**e**) *V_oc_* for as-fabricated PS-2 cell, passivated PS-2-(a-h) cells with different *t*, and reference *c*-Si cell.

**Figure 7 nanomaterials-11-00459-f007:**
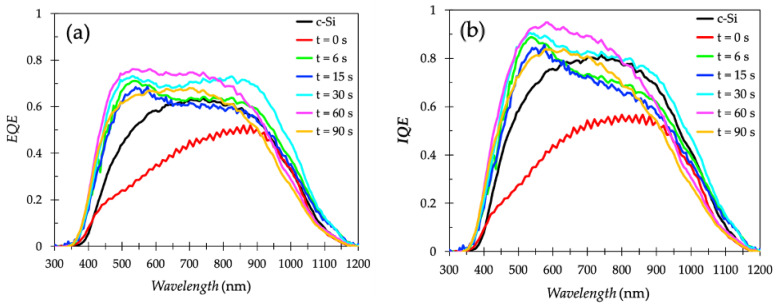
(**a**) EQE; (**b**) IQE of PS-2 cells for different *t* compared with as-fabricated PS-2 cell and reference *c*-Si cell.

**Figure 8 nanomaterials-11-00459-f008:**
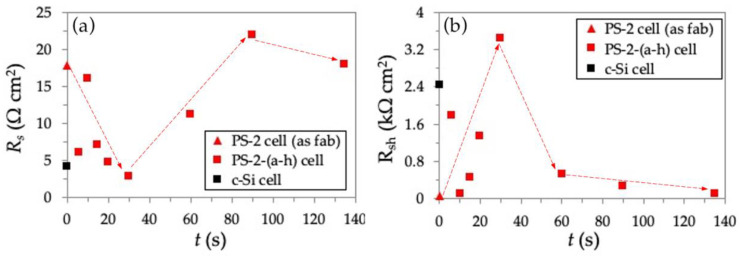
Correlation between *t* and (**a**) *R_s_*; (**b**) *R_sh_* for as-fabricated PS-2 cell, passivated PS-2(a-h) cells, and reference *c*-Si cell.

**Table 1 nanomaterials-11-00459-t001:** Overview of preparation conditions and experiments for all samples.

Samples	Current Density (mA/cm^2^)	Etching Duration (s)	Passivation Duration (s)	PS Layer Thickness (nm)	Volume Ratio of SiO_2_ to PS Layer (%)	Experiment	Section
Samples A,B,C	0	0	0	0	0	ECV 5 mm^2^	Results and Discussion part 3.1
Sample D	0	0	0	0	0	ECV 0.7 mm^2^	
PS-1 layer	75	6	0	146	0	TEM HRTEM	
PS-2 layer	75	6	0	176	0		
PS-3 layer	① 75 ② 5	① 10 ② 300	0	545	0		
*c*-Si cell	0	0	0	0	0	Reflectivity *J–V* curve EQE	Results and Discussion part 3.2
As-fabricated PS-1 cell	75	6	0	150	0		
Passivated PS-1 cell	75	6	-	-	-		
As-fabricated PS-2 cell	75	6	0	191	0	FESEM Reflectivity *J–V* curve EQE IQE	Results and Discussion part 3.3
Passivated PS-2-a cell	75	6	6	239	0.6		
Passivated PS-2-b cell	75	6	10	224	0.9		
Passivated PS-2-c cell	75	6	15	205	1.4		
Passivated PS-2-d cell	75	6	20	255	1.9		
Passivated PS-2-e cell	75	6	30	235	2.8		
Passivated PS-2-f cell	75	6	60	216	5.6		
Passivated PS-2-g cell	75	6	90	213	8.5		
Passivated PS-2-h cell	75	6	135	210	12.7		

[-: data not available].

**Table 2 nanomaterials-11-00459-t002:** PV parameters for PS-1 cell, as-fabricated passivated PS-1 cell, and reference *c*-Si cell.

	*c*-Si Cell	As-Fabricated PS-1 Cell	Passivated PS-1 Cell
*J_sc_* (mA/cm^2^) ^1^	23.7	27.5	30.5
*V_oc_* (V)	0.510	0.510	0.510
*FF*	0.660	0.300	0.550
*η* (%) ^1^	7.93	4.15	8.54

^1^*J_sc_* and *η* were calculated based on the active area.

**Table 3 nanomaterials-11-00459-t003:** PV parameters for as-fabricated PS-2 cell, passivated PS-2 cells with different *t*, and reference *c*-Si cell.

Samples	*t* (s)	PS Layer Thickness (nm)	*V_oc_* (V)	*J_sc_* (mA/cm^2^) ^1^	*FF*	*η* (%) ^1^
*c*-Si cell	0	0	0.532	23.4	0.657	8.16
As fabricated PS-2 cell	0	191	0.539	29.9	0.320	5.15
Passivated PS-2-a cell	6	239	0.532	27.9	0.530	7.86
Passivated PS-2-b cell	10	224	0.524	27.6	0.364	5.25
Passivated PS-2-c cell	15	205	0.514	29.6	0.466	7.08
Passivated PS-2-d cell	20	255	0.531	30.4	0.578	9.32
Passivated PS-2-e cell	30	235	0.523	29.8	0.685	10.70
Passivated PS-2-f cell	60	216	0.526	28.7	0.462	6.97
Passivated PS-2-g cell	90	213	0.513	26.2	0.357	4.79
Passivated PS-2-h cell	135	210	0.511	25.0	0.354	4.52

^1^*J_sc_* and *η* were calculated based on the active area.

## Data Availability

Data is contained within the article or Appendix A.

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
