# Peer review of "Nanostructure of Porous Si and Anodic SiO2 Surface Passivation for Improved Efficiency Porous Si Solar Cells"

_nanomaterials, 2021, doi:10.3390/nano11020459_

Round 1

Reviewer 1 Report

The paper deals with silicon solar cells fabricated by using porous silicon as a light absorber layer in the emitter region of the cell. The high surface to volume ratio increases the surface traps. To passivate such defects, the authors perform anodic oxidation. They changed the thickness of the oxide layer and characterized the samples by HRTEM and extensive electrical measurements.

The main weakness of the manuscript is related to the real novelty of this approach. The anodic oxidation has already been demonstrated in 2019 [N.E. Grant, T.C. Kho, K.C. Fong, E. Franklin, K.R. McIntosh, M. Stocks, Y. Wan, Er-Chien Wang, N. Zin, J.D. Murphy, A. Blakers, Anodic oxidations: Excellent process durability and surface passivation for high efficiency silicon solar cells, Solar Energy Materials and Solar Cells, Volume 203, 110155 (2019)]. So, it is not clear what the real contribution of this paper is, with respect to the literature.

Nevertheless, if the authors can explicitly state what is the advance with respect to the state of the art, commenting it properly, in the abstract, introduction and in the discussion of the data, maybe the novelty will become clear to the audience and the paper will give a contribution.

Reviewer 2 Report

The entitled manuscript ‘Nanostructure of Porous Si and Anodic SiO2 Surface Passivation for Improved Efficiency Porous Si Solar Cells’ by Panus Sundarapura et. Al is describing about the effect of anodic formation of silicon dioxide (SiO2) on porous silicon (PS) on the photovoltaic performance. It can be considered to be published in MDPI nanomateirals after major revision process.

  • First of all, I could not find the strength of the suggested concept in the points of view, science, engineering as well as from an economic. Porous nanostructure on Si is already reported several times and suggested method is also not suitable to actual production process.
  • The authors argue that bottom-up anodization based SiO2 passivation layer on PS can improve the efficiency of Si PV and the uniformity of SiO2 passivation layer. But I think the tested sample is too much small size of active layer to evaluate the ideal concept with low-cost processing that is one of the import argues pointed out by the authors several times. Don’t you try with wafer scale? I would like to suggest to reporting scalable PV results.
  • For Ag electrode, is it too much thin for the front and rear surface electrode? is there any reason? and do not make BSF?
  • The suggested reference Si cell efficiency is too much low.

Reviewer 3 Report

Manuscript presents the interesting information of the developing of nanostructured SiO2 layer for improving the performance of Si solar cells. Paper can be considerate for publication after minor revision and correction of the English in abstract. Also additional discussion should be provided regarding the improvement of the discussion part.

Comments:

  1. English:

Line 18. “anodic layer was passivated”. Sounds weird. “Passivation” implies using passivation by something. The layer on some substrate can be “created” or “deposited”.

Line 25. “suitable anodization duration of 30 s achieved the best conversion efficiency”. Sentence in incorrect. How can duration of anodization can be achieved? “Determined” is more appropriate verb here. 2nd part of sentence: how “anodization duration” can achieve “energy conversion efficiency”?

  1. Fig. 1a. Please adjust text on the Figure for better visibility. Text should not be overlapped with lines of schematics (Pt net).
  2. Line 116. Please comment on how the porosity of 38% was estimated.
  3. Regarding the paper organization the definition of samples should be give in more clear way. For example, on line 153 and Fig. 2 samples A-D are mentioned. But it is hard to find the explanation on what are those samples. This point should be explained in more clear way.
  4. Fig. 2a. Please comment why sample B shows such a high carrier concentration at surface.
  5. Fig. 3. Authors distinguish “pore” area in TEM image, however from image “pore” and “crystalline Si” areas look very similar. Additional analysis of porosity of PS layer can be provided to improve paper.
  6. The confirmation of formation of SiO2 layer during anodization process should be given.

Round 2

Reviewer 2 Report

I have carefully read again this revised version.

All my concerns were properly addressed.

I think it can be acceptable for publicaction.